# Effect of Crack Defects on Magnetostriction and Magnetic Moment Evolution of Iron Thin Films

**DOI:** 10.3390/nano12071236

**Published:** 2022-04-06

**Authors:** Hongwei Yang, Meng Zhang, Lianchun Long

**Affiliations:** 1Faculty of Science, Beijing University of Technology, Beijing 100124, China; mel98ray99@163.com; 2Faculty of Materials and Manufacturing, Beijing University of Technology, Beijing 100124, China; longlc@bjut.edu.cn

**Keywords:** iron, thin film, magnetostriction, crack defects, molecular dynamics

## Abstract

Molecular dynamics simulations of body-centered cubic (bcc) iron thin films with crack defects were carried out by adopting methods of EAM (Embedded Atom Method) potential, spin/exchange potential and spin/neel potential. In this article, the effects of the variation of distance between two crack defects and their directions on the magnetostrictive properties of the thin films are studied, and the corresponding microscopic mechanism is also analyzed. The results show that the defects affect the atomic magnetic moment nearby, and the magnetostrictive properties of thin iron films vary with the direction and spacing of the crack defects. If the defect spacing is constant, the iron model with crack perpendicular to the magnetization direction has stronger magnetostriction than that of parallel to the magnetization direction. The variation of the defect spacing has a great influence on the magnetostrictive properties of the iron model with crack direction parallel to magnetization direction, but it has a small effect on another perpendicular situation. The atoms between the defects may move, but if the defect spacing increases to a certain value, then none of the atoms will move.

## 1. Introduction

When a ferromagnetic material is exposed to a magnetic field, its dimensions change, and this effect is called magnetostriction [1,2,3,4]. Due to the excellent characteristics of large magnetoelastic coupling coefficient, large output stress, fast mechanical response and strong stability, magnetostrictive materials have important application value and broad application prospects in the fields of robot, automobile brake, sensor, transducer, displacement device, high-energy micro power device, acoustics, magnetism and so on [5,6,7,8].

At present, most magnetostrictive materials are made by alloy or doping based on iron, such as Terfenol-D (TbDyFe alloy) and Fe-Ga alloy [9,10,11,12]. TbDyFe alloy has high magnetostriction but weak tensile and brittle texture. Polycrystalline Fe-Ga alloy can be easily prepared and has a low price but a low magnetostrictive coefficient [13]. Therefore, the research on magnetostriction of iron has certain significance to improve the properties of magnetostrictive materials.

Temperature, stress and defects all affect the magnetostrictive properties of materials [14]. Scholars have achieved many research results on the influence of temperature and stress on magnetostriction [15,16]. The studies of defects have mostly focused on detecting whether defects exist [17]. In this regard, scholars have conducted a series of theoretical and experimental studies [18,19], but the effects of defects on magnetostrictive materials and their internal mechanisms are still not discussed in detail.

As for the research on the effect of defects on magnetostrictive materials, Zhang Shuo et al. [20] established an iron magnetostrictive structure model without defects, with a hole defect and with a crack defect, respectively, and analyzed the influence of defect form on the magnetostrictive behavior of iron thin film. Based on the above work, an iron model with two crack defects is established in this article by the molecular dynamics method. The effects of the distance between two crack defects and their directions on the magnetostrictive properties of bcc iron films are studied, and the influence mechanism on the change of internal microstructure is analyzed from the level of atomic magnetic moment.

## 2. Calculation Method

Molecular dynamics is a simulation method to describe the laws of atomic motion in real molecular system. In this paper, the molecular dynamics method is used to analyze the effect of crack defects on magnetostriction and magnetization configuration evolution of iron thin films under different applied magnetic fields. In the simulation, since the magnetic interaction has an effect on the calculation results, the exchange interactions between the spin pairs and the anisotropy of the magnetic crystals are taken into account in the choice of potential functions. The interatomic potential is simulated by EAM (Embedded Atom Method) potential, spin/exchange potential [21,22,23] and spin/neel potential [21], using LAMMPS (Large-scale Atomic/Molecular Massively Parallel Simulator) molecular dynamics software. Spin/exchange potential is used to calculate the exchange interaction between spin pairs; spin/neel potential is used to describe the anisotropy of the magnetic crystals.

The original size of the model is set to *N_x_* × *N_y_* × *N_z_*, where *N_x_*, *N_y_* and *N_z_* represent the lattice number along the *x*, *y* and *z* cell directions, respectively, the default orientation of *x*, *y* and *z* is [1 0 0], [0 1 0] and [0 0 1] crystal directions. Here, the simulated size is set to 160 × 160 × 1 lattice periods, and the lattice constant is 2.86 Å. Then, by removing 2 groups of 60 × 2 lattice atoms in the center of the model and adjusting the distance between the two groups defects to 2, 4, 6 and 8 lattice periods, respectively, four 60 × 2 × 2 defects models with different defect spacing are constructed; by deleting 2 groups of 2 × 60 lattice atoms in the center of the model and adjusting the distance between the two groups defects to 2, 4, 6, and 10 lattice periods, respectively, four 2 × 60 × 2 defects models with different defect spacing are constructed. 

The boundary condition of the model is SSP (S: non-periodic and shrink-wrapped, P: periodic), i.e., non-periodic boundary condition in the *x*, *y* direction, and periodic boundary condition in the *z* direction. The atomic magnetic moment of iron atom is 2.2 *μ_B_* (Bohr magneton), and the initial magnetic moment direction of the atom is set as a random distribution to ensure that the overall magnetization of the model is approximately close to 0. The above models are simulated at 300 K with nve (microcanonical ensemble)/spin ensemble and Langevin/spin temperature controller. The integral step is 5 fs, the data output interval is 50 fs, and the relaxation is 300 ps to ensure the equilibrium of the models.

## 3. Results and Discussion

### 3.1. The Models with 60 × 2 × 2 Defects

After the relaxation equilibrium of the models with 60 × 2 × 2 defects, the output models are considered to be the initial models. The magnetic moment of each atom in the initial models is projected onto the *xy* plane to obtain the magnetization configuration diagrams, as shown in Figure 1. The directions of the arrows in the figure represent the direction of the magnetic moment. In order to observe atoms’ magnetized degree, we calculate the average value ***u*** of all normalized atomic magnetic moments. For example, the component of ***u*** in the *x* direction is given by
(1)$$\begin{array}{c} u_{x}\;=\;\frac{\sum_{i\;=\;1}^{n}\frac{l_{xi}}{\sqrt{l_{xi}^{2}\;+\;l_{yi}^{2}\;+\;l_{zi}^{2}}}}{n} \end{array}$$
where *n* is the total number of atoms for the model, and *l_x_*, *l_y_*, *l_z_* are the components of the atomic magnetic moment in the *x*, *y*, *z* directions, respectively.

It can be seen from Figure 1 that the initial magnetization configuration diagrams with a defect spacing of 2, 6 and 8 lattice periods have similar performance. Their overall magnetizations are still about 0. Due to the existence of defects, two trapezoidal regions are formed above and below the defects, in which the atomic magnetic moments are in the positive and negative *x*-axis, respectively; two triangular regions are formed on the left and right parts, in which the atomic magnetic moments are in the positive and negative *y*-axis, respectively; and their magnetic moments between the four regions are in transition direction. For the model where the distance is 2, the atoms between the two defects move during the relaxation process, so the two defects become one defect. After the relaxation of the models where the distances are 6 and 8, their atomic magnetic moments between the two defects are in the negative *x*-axis direction. In the above three models, *u_x_* is about −0.03, and *u_y_* is around −0.04. For the model with the distance between the two defects to be 4 lattice periods, the direction of the atomic magnetic moments between the two defects is, from left to right, along the positive, negative and positive *x*-axis. At this moment, *u_x_* is 0.75, and *u_y_* is 0.51. Its overall magnetization is no longer 0 after the relaxation, and its initial magnetization configuration is different from the above three situations. When the defect spacing is 20 lattice periods, a similar situation occurs, and its overall magnetization configuration is not 0.

Figure 2 shows the result of the magnetostrictive fitting curves by magnetizing the above initial models along the positive direction of *x*-axis (the magnetization direction is parallel to the crack direction) and then changing the magnetic field strength to obtain the strain data. The final strain values are the average values of stabilized data. In this figure, the abscissa is the normalized magnetic field intensity, *H* is the magnetic field strength, and *H_m_* is the maximum external magnetic field that saturates the model with magnetostriction.

Figure 2a illustrates that the magnetostrictive strains of the five models in the *x* direction all increase with the addition of magnetic field. The magnetostrictive strain of the four defect models is smaller than that of the no defect model. By comparing the four defect models, it can be seen that the models where the distances are 2 and 4 need a larger magnetic field to approach saturation, and the former has the largest saturation magnetostriction with a value of 0.45%; the model that the distance is 8 is the easiest to achieve saturated magnetostriction, and its saturated magnetostriction is the smallest with a value of 0.21%; the saturated magnetostrictive strains of the four models span nearly 0.25%. As can be seen from Figure 2b, in the *y* direction, when the distance is 4, the shrinkage strain is the smallest; while if the distance is 2, the shrinkage strain is the largest.

In order to further explore the influence of distance between defects on magnetostrictive behavior from the microscopic level, the magnetization configuration diagrams of four models with the change of atomic magnetic moment are made under the magnetization of different magnetic fields. 

Figure 3 shows the evolution of the magnetization configuration of the model; the distance between defects is 2 lattice periods under different magnetic fields. By the action of the 0.025 *H_m_* magnetic field, the atomic magnetic moments under the defects originally pointing to the negative *x*-axis change to the positive direction of *x*-axis. At the upper boundary of the model, an anti-magnetization vortex and a magnetization vortex appear due to the existence of defects. At this moment, *u_x_* changes from −0.02 to 0.62, which is a significant change, and *u_y_* is 0.07. When the applied magnetic field is 0.375 *H_m_*, the magnetization of the model begins to approach saturation. The atomic magnetic moments at the upper and lower regions of the defects are in the positive direction of the *x*-axis, and at the left and right parts of the defects, the magnetic moments are in the transition direction. When the applied magnetic field increases to 1 *H_m_*, the model reaches saturation. The defect is surrounded by the atomic magnetic moments along the positive direction of the *x*-axis, and only part of the atomic magnetic moments at the left and right boundaries have not completely turned to the positive direction of the *x*-axis.

Figure 4 illustrates a situation when the defect spacing is 4 lattice periods. In the magnetic field of 0.025 *H_m_*, the atoms between the two defects move, which resulting in the two defects becoming one defect. The atomic magnetic moments in the transition region are decreasing, while along the positive *x*-axis are increasing. At this moment, *u_x_* changes slightly from 0.75 to 0.81, and *u_y_* is 0.003. When the applied magnetic field is 0.375 *H_m_*, the model approaches saturation, and the two defects also become one defect. Under the action of a strong magnetic field 1 *H_m_*, the model reaches saturation, and the atoms between the defects also move. Similarly, only the atomic magnetic moments of the left and right boundaries have not completely shifted to the positive direction of the *x*-axis.

Figure 5 shows a situation where the defect spacing is 6 lattice periods. Under the action of a 0.025 *H_m_* magnetic field, the atomic magnetic moments between the two defects are in the positive *x*-axis. The four regions with different atomic magnetic moments and the transition region have certain displacement and deformation, but the area occupied by each region does not change dramatically. At this moment, *u_x_* is 0.17 and *u_y_* is −0.24. Under the magnetization of the 0.375 *H_m_* magnetic field, the model is also close to saturation, and the direction of atomic magnetic moments between the two defects is still in the positive direction of the *x*-axis. Compared with the above two models, the area of the transition region becomes larger. When the applied magnetic field increases to 1 *H_m_*, the model reaches saturation, and the two defects become one defect. Compared with the above two models, only the area of the atomic magnetic moments in the transition direction becomes larger.

Figure 6 displays a situation where the defect spacing is 8 lattice periods. The atoms between the two defects no longer move by the action of the magnetic field. In the magnetic field of 0.025 *H_m_*, *u_x_* is 0.02, and *u_y_* is 0.03, with the minimum change. When the applied magnetic field is 0.375 *H_m_*, the model is close to saturation. The atomic magnetic moments between the two defects follow the positive, negative and positive *x*-axis directions from left to right, respectively. Because the distance between the two defects is too large, the atomic magnetic moments of the upper and lower parts of the defects do not all follow the positive direction of the *x*-axis. Under the magnetization of an external 1 *H_m_* magnetic field, the model reaches saturation. The defects are surrounded by the atomic magnetic moments along the positive direction of the *x*-axis. Compared with the above three models, the area of the atomic magnetic moments in the transition direction at the left and right boundaries of this model is the largest.

### 3.2. The Models with 2 × 60 × 2 Defects

After the models with 2 × 60 × 2 defects are relaxed, the output model is considered to be the initial model, and the magnetic moment of each atom in the initial models is projected onto the *xy* plane to obtain the magnetization configuration diagrams, as shown in Figure 7.

It can be seen from Figure 7 that the initial models with a defect spacing of 2, 4 and 6 lattice periods have similar performance. The atomic magnetic moments in the middle region of the models are positive along the *y*-axis, at the upper and lower boundaries are positive along the *x*-axis, and between the above regions are in the transition direction. The difference among these three models is that with the increasing of the distance, the atomic magnetic moment in the positive direction of the *y*-axis decreases, while in the transition direction increases. In the model where the distance is 2, the atoms in the middle of two defects move in the relaxation process, so the two defects merge into one defect. For the model where the distances are 4 and 6, the atomic magnetic moments between the two defects are positive along the *y*-axis. For the model where the distance is 10 lattice periods, the direction of atomic magnetic moments on the left side of the defects is close to the positive direction of *y*-axis, and the direction of the remaining regions is roughly similar to the corresponding regions of the above three models. Compared with Figure 1, as in Figure 1a and Figure 7a, after the rotation by 90 degrees, the initial magnetized configuration is inconsistent. The reason is that the simulation calculations take the interaction between spin pairs into account, and it causes differences in the *x*, *y* direction so that there is no coincidence after 90 degrees of rotation.

Figure 8 displays the result of the magnetostrictive strain curves in the *x**, y* directions by magnetizing the above initial models along the positive direction of *x*-axis (the magnetization direction is perpendicular to the crack direction).

Figure 8a shows, in the *x* direction, the magnetostrictive strain of the four defect models is larger than that of the no defect model. The model with the distance of 4 lattice periods requires a large magnetic field to approach saturation, and its saturation magnetostriction is the largest with a value of 0.64%; the saturation magnetostrictions of the other three defect models are similar, with a value of 0.59%; the saturated magnetostrictive strain of the four defect models only changed by 0.05%, which is compared with 0.25% of the 60 × 2 × 2 defect model, so the defect spacing has a small effect on the magnetostrictive strain of the 2 × 60 × 2 defect model. As shown in Figure 8b, in the *y* direction, for the model with 2 lattice periods between the defects, the shrinkage strain is the largest when comparing the four defect models; for the models in which the defect spacings are 4 and 6, their shrinkage strains are close to each other; for the model where the distance is 10, the shrinkage strain is the smallest.

The magnetization configuration diagrams of the above four models after reaching equilibrium under magnetization of different magnetic fields is studied below.

Figure 9 illustrates, under different magnetic fields, the evolution of the magnetization configuration of the model where the distance between defects is 2 lattice periods. With the magnetization of the 0.025 *H_m_* magnetic field, the region of atomic magnetic moments along the positive direction of *y*-axis decreases greatly. When the applied magnetic field is 0.375 *H_m_*, the magnetostriction begins to approach saturation, the atomic magnetic moments at the left and right boundaries of the model and around the defects are in the transition direction. Under the impact of a strong magnetic field 1 *H_m_*, the model reaches saturation. The magnetic moments of some atoms around the defects are in the transition direction, and their moments at the left and right boundaries of the model have not completely turned to the positive direction of the *x*-axis.

Figure 10 shows a situation where the defect spacing is 4 lattice periods. With the magnetization of the 0.025 *H_m_* magnetic field, the regions with different atomic magnetic moments rotate counterclockwise around the defects, and the atomic magnetic moments between the two defects are still along the positive direction of *y*-axis. Under the action of the 0.375 *H_m_* magnetic field, the model is also close to saturation. The atoms between the two defects start to move, causing two defects to become one defect. The defect is surrounded by the atomic magnetic moments which are in the transition direction. The atomic magnetic moments in the right area of the defect are close to the positive direction of the *x*-axis, and their moments at the left and right boundaries are in the transition direction. When the applied magnetic field increases to 1 *H_m_*, the model reaches saturation and the movement of the atoms between the two defects also occurs. There are still some atoms either near the left and the right boundaries of the model or around the defects whose magnetic moments are along the transition direction.

Figure 11 displays a situation when the defect spacing is 6 lattice periods. Under the magnetization of the 0.025 *H_m_* magnetic field, the region along the positive direction of *x*-axis gradually increases. The atoms between the two defects move, and the two defects become one defect. However, when the defect spacing is 4, the two defects do not merge into one, as shown in Figure 10a. This shows that it is not that the smaller the defect spacing is, the easier that the merging of the two defects happens. It probably indicates that the phenomenon of defects merging is not only related to the distance of defects. When the applied magnetic field is 0.375 *H_m_*, the magnetostriction begins to approach saturation, and the atoms between the two defects also start to move. There are some atoms not just around the defects but also at the middle of the model whose magnetic moments are in the transition direction. Under the action of a strong magnetic field 1 *H_m_*, the model reaches saturation. The atomic magnetic moments around the defects and at the left and right boundaries of the model have not completely turned to the positive direction of the *x*-axis.

Figure 12 shows a situation where the defect spacing is 10 lattice periods. No matter how large the magnetic field is, the atoms between the two defects do not move again. In the magnetic field of 0.025 *H_m_*, the area of the atomic magnetic moments along the *x*-axis begins to increase. When the applied magnetic field is 0.375 *H_m_*, the model is also close to saturation. The atomic magnetic moments between the two defects no longer follow the positive direction of the *y*-axis but become close to the negative direction of the *y*-axis. Under the action of a strong magnetic field 1 *H_m_*, the model reaches saturation. Either at the left and right boundaries of the model or around the defects, the area of the atomic magnetic moments in the transition direction becomes larger than in the previous three models.

## 4. Conclusions

A model of bcc iron thin film with crack defects is established using the molecular dynamics method. The influence of different crack defects spacing on the magnetostrictive behavior of iron thin films was studied, and the changes of internal microstructure were also analyzed. The following conclusions were drawn:(1)The direction of defects and the defect spacing both affect the magnetostrictive properties of iron thin films.(2)If the defect spacing is constant, the iron model with a crack direction perpendicular to the magnetization direction has a higher magnetostriction than the iron model with a crack direction parallel to the magnetization direction. When the crack direction in the component is parallel to the magnetization direction, the atoms around the crack are easier to magnetize compared to when the crack direction is perpendicular in direction, which will result in a decrease of the maximum magnetostriction from the initial state to saturation in the magnetization direction.(3)The change of the distance between defects has a great influence on the magnetostrictive properties of the iron thin film model with the crack direction parallel to the magnetization direction, and it has a small influence on the magnetostrictive properties of the iron thin film model with the crack direction perpendicular to the magnetization direction.(4)When the distance between defects is small, in order to reach a more stable state under the magnetization of the magnetic field, the atoms between defects may move, resulting in two defects becoming one defect. When the distance between defects increases to a certain value, the atoms between defects will no longer move under any magnetic field.(5)Under the action of a strong magnetic field, both the iron model with crack direction parallel to the magnetization direction and the iron model with crack direction perpendicular to the magnetization direction reach saturation. In the former, only part of the atomic magnetic moment at the left and right boundaries of the model does not turn in the direction of the magnetization; in the latter, not only at the left and right boundaries of the model but also around the defect, the magnetic moment of some atoms does not turn in the direction of the magnetization.

## Figures and Tables

**Figure 1 nanomaterials-12-01236-f001:**
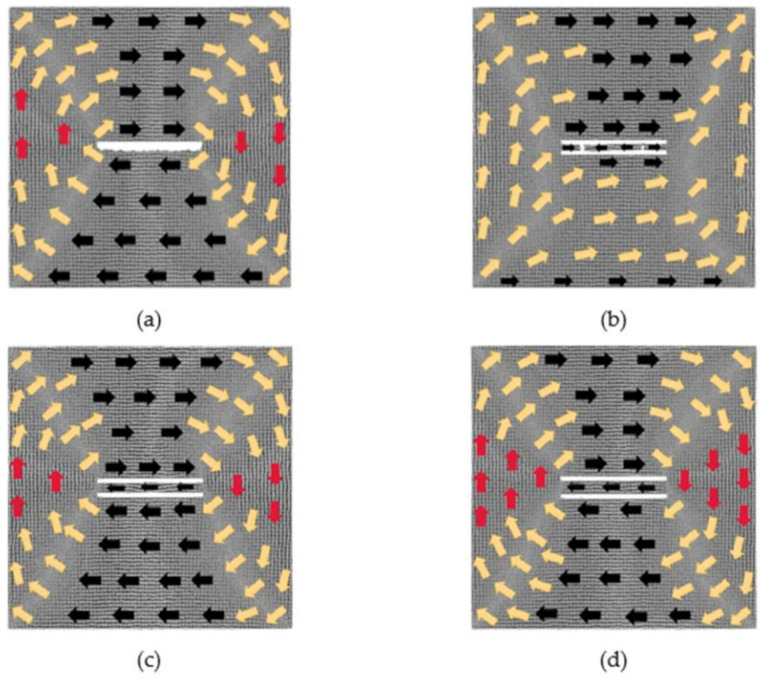
Initial magnetization configuration diagrams of models with 60 × 2 × 2 defects in the center: (**a**) 2 lattice periods, (**b**) 4 lattice periods, (**c**) 6 lattice periods and (**d**) 8 lattice periods.

**Figure 2 nanomaterials-12-01236-f002:**
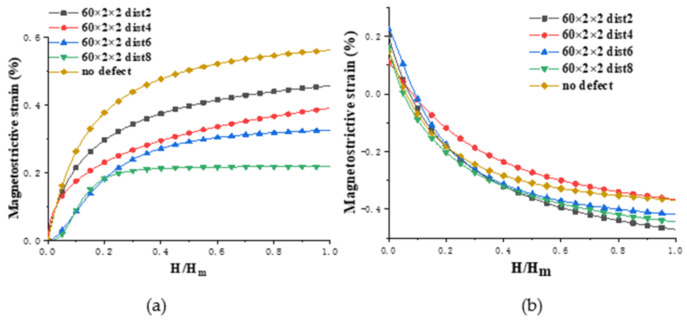
Comparison of the magnetostrictive strains of 60 × 2 × 2 defects models and no defect model: (**a**) *ε**_xx_* and (**b**) *ε**_xy_* (the labels *xx* and *xy* denote the magnetostriction measured in the *x* and *y* directions, respectively, when a magnetic field is applied in the *x* direction).

**Figure 3 nanomaterials-12-01236-f003:**
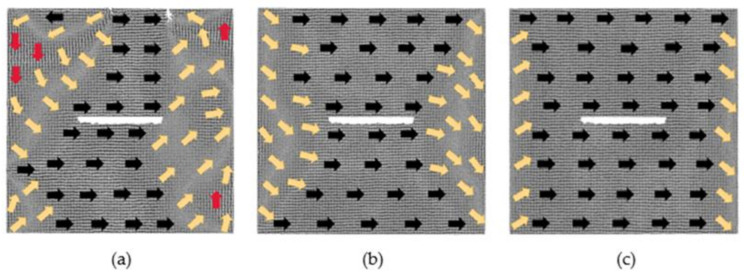
Evolution diagrams of magnetization configuration of model with 60 × 2 × 2 defects in the center (2 lattice periods): (**a**) 0.025 *Hm*, (**b**) 0.375 *Hm* and (**c**) 1 *Hm*.

**Figure 4 nanomaterials-12-01236-f004:**
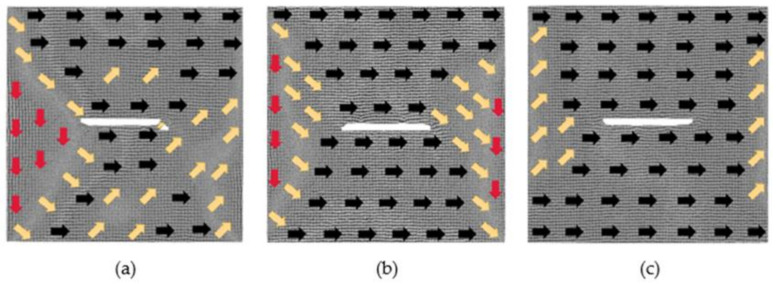
Evolution diagrams of magnetization configuration of model with 60 × 2 × 2 defects in the center (4 lattice periods): (**a**) 0.025 *H_m_*, (**b**) 0.375 *H_m_* and (**c**) 1 *H_m_*.

**Figure 5 nanomaterials-12-01236-f005:**
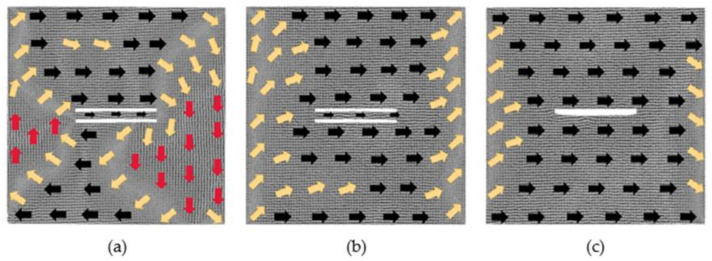
Evolution diagrams of magnetization configuration of model with 60 × 2 × 2 defects in the center (6 lattice periods): (**a**) 0.025 *H_m_*, (**b**) 0.375 *H_m_* and (**c**) 1 *H_m_*.

**Figure 6 nanomaterials-12-01236-f006:**
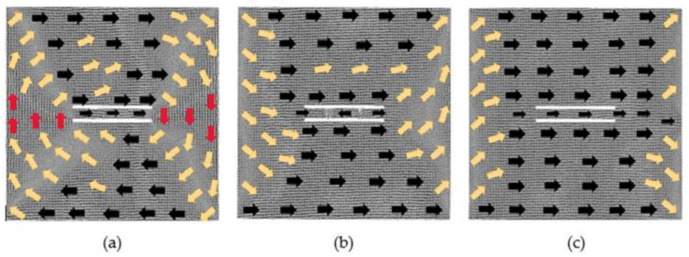
Evolution diagrams of magnetization configuration of model with 60 × 2 × 2 defects in the center (8 lattice periods): (**a**) 0.025 *H_m_*, (**b**) 0.375 *H_m_* and (**c**) 1 *H_m_*.

**Figure 7 nanomaterials-12-01236-f007:**
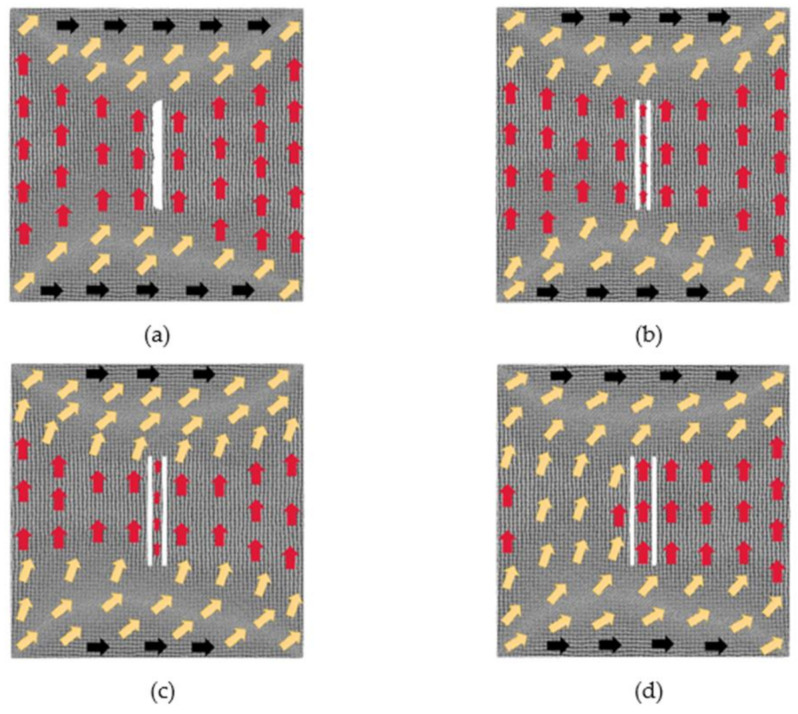
Initial magnetization configuration diagrams of models with 2 × 60 × 2 defects in the center: (**a**) 2 lattice periods, (**b**) 4 lattice periods, (**c**) 6 lattice periods and (**d**) 10 lattice periods.

**Figure 8 nanomaterials-12-01236-f008:**
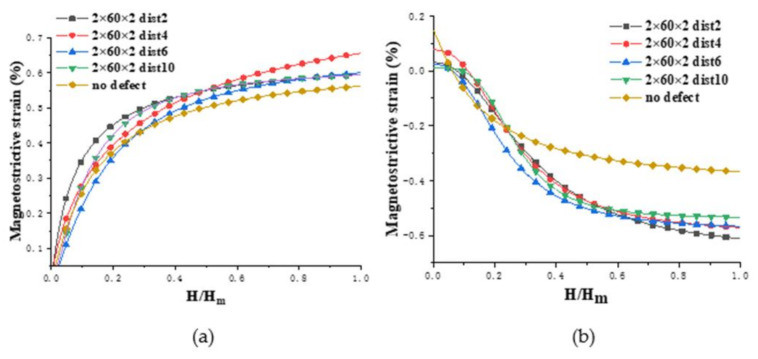
Comparison of the magnetostrictive strains of 2 × 60 × 2 defects models and no defect model: (**a**) *ε**_xx_* and (**b**) *ε**_xy_*.

**Figure 9 nanomaterials-12-01236-f009:**
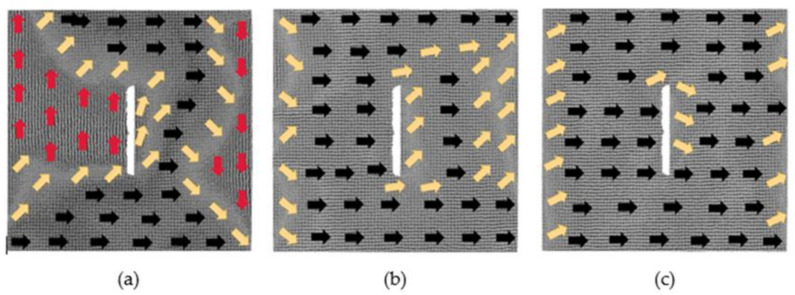
Evolution diagrams of magnetization configuration of model with 2 × 60 × 2 defects in the center (2 lattice periods): (**a**) 0.025 *H_m_*, (**b**) 0.375 *H_m_* and (**c**) 1 *H_m_*.

**Figure 10 nanomaterials-12-01236-f010:**
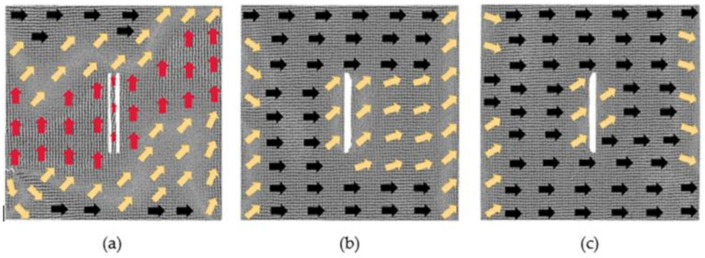
Evolution diagrams of magnetization configuration of model with 2 × 60 × 2 defects in the center (4 lattice periods). (**a**) 0.025 *H_m_*, (**b**) 0.375 *H_m_*, (**c**) 1 *H_m_*.

**Figure 11 nanomaterials-12-01236-f011:**
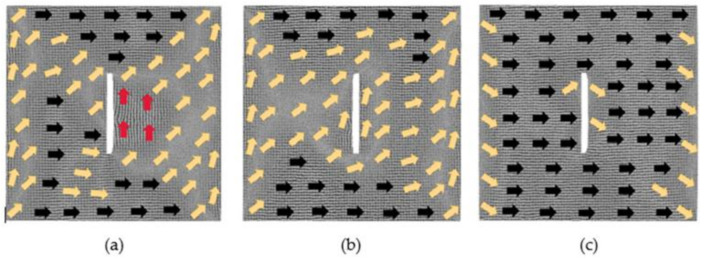
Evolution diagrams of magnetization configuration of model with 2 × 60 × 2 defects in the center (6 lattice periods): (**a**) 0.025 *H_m_*, (**b**) 0.375 *H_m_* and (**c**) 1 *H_m_*.

**Figure 12 nanomaterials-12-01236-f012:**
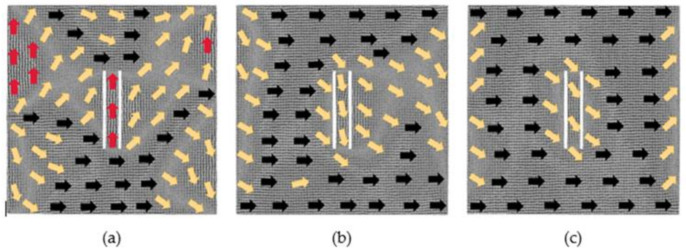
Evolution diagrams of magnetization configuration of model with 2 × 60 × 2 defects in the center (10 lattice periods): (**a**) 0.025 *H_m_*, (**b**) 0.375 *H_m_* and (**c**) 1 *H_m_*.

## Data Availability

The data that support the findings of this study are available from the corresponding author upon reasonable request.

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
