# Peer review of "Effect of Crack Defects on Magnetostriction and Magnetic Moment Evolution of Iron Thin Films"

_nanomaterials, 2022, doi:10.3390/nano12071236_

Round 1

Reviewer 1 Report

The article presents the results of modeling magnetostriction and micromagnetic structure in a monoatomic layer of iron with dimensions of 160x160 atoms (458x458 Angstroms) in the presence of two model cracks. The dimensions of each crack were assumed to be equal to 60x2 atoms. Calculations were carried out by the method of molecular dynamics in the LAMMPS software package. The magnetic moment of a single atom was assumed to be equal to 2.2 Boron magnetons. The cracks were located parallel to each other in the center of the sample. Two cases of the location of cracks relative to the magnetizing field are considered - along and perpendicular to the direction of the magnetic field.

The authors analyze the changes in the distributions of magnetic moments in the samples when the magnetic field strength changes and consider the behavior of magnetostrictive stresses when the distance between cracks changes from two lattice periods to eight (ten in the case of the orientation of cracks perpendicular to the magnetic field).  The conclusions give a brief summary of the described results.

When reading the text of the article, many questions arise.

How was the preservation of the sample plane ensured in the absence of a substrate? Without a substrate, the atoms would form a spherical cluster if the calculations were carried out correctly.

The article does not say how the magnitude of the magnetostrictive strain shown in Graphs 2 and 8 was estimated.

How the susceptibility was calculated?

The description of the model says that the samples were square, and the figures show rectangles. Why?

The initial configuration of the distribution of magnetic moments in Figures 1 and 7 should coincide, taking into account the rotation by 90 degrees, but Figures 1 (a) and 7 (a), 1 (b) and 7 (b), etc. do not coincide, why?

What will happen with magnetostrictive strain in case of the cracks shifting to the sample boundary?

How do the dimensions of the arrows indicating the orientation of the magnetic moments in the figures correlate with the magnitude of the magnetic moments?

How did the movement of atoms occur when two defects merged into one? Why is there no merging at a distance of 4 periods in the 0.025 Hm field (Fig. 10), and at a distance of 6 periods it occurs (Fig. 11) in the same field?

In addition, the English language should be significantly improved.
The first sentence on the introduction is wrong, the magnetostriction appears in FERROMAGNETIC materials when the magnetization changes. (line 24-25). There is no expressions neither "metal magnetic memory detection" (line 39) no "the magnetic memory signal" in English language (line 42). Instead of "lattice size" should be used "lattice period" (in all text). The abbreviation must be described (lines 59, 60, 71, 75 and so on).

In present form the manuscript can't be recommended for publication.

Author Response

Dear reviewer:

We appreciate the time and effort that you dedicated to providing feedback on our paper entitled “Effect of crack defects on magnetostriction and magnetic moment evolution of iron thin films (Nanomaterials-1641696)”. We have made corresponding revisions according to your advice. My response to your questions and suggestions are listed as follows.

Reviewer 2 Report

The paper describes dependence of magnitization on presence of defects. This paper is important because of huge application potential of magnetostrictive materials.

Anyway, it should be corrected before publishing, because it contains misprints (all abbreviations should be described too). It also should be brought to higher English level. Please, put the description of figures’ parts (a,b,c,…) to the captures of these figures.

This sentence can be supported by references:  Temperature, stress and defects all affect the magnetostrictive properties of materials [DOI: 10.3390/NANO10101990].

Please, mention why the Molecular dynamics is the best choice in your case.

I also miss explanation of the chosen defects models.

Author Response

(The authors gave the same response as above.)

Reviewer 3 Report

The paper refers to magnestostrictive processes of defected Fe structure. The approach and results are very interesting and investigations worthy. The Authors presented also magnetization processes and spin structures under increasing external magnetic field. After some corrections the paper has a potential to be helpful for researchers working in the field of magnetostrictive materials, therefore, I recommend the paper for publication but after a review.

My suggestions and comments:

1. A 3D graph of the studied system, with the supercell and repetition directions, should be included. Explain, why the periodic conditions were established for z and not for x,y that is expected in thin layer models. If I well understand, the Authors analyzed infinite nano-rod, not thin layer.

2. The model of magnetic interactions (dipolar exchange, magnetostatic) should be better explained (only references are too weak). Where is magnetic anaisotropy (shape and magnetocrystalline types are important) taken into consideration? Results can significantly vary accordingly to used model of magnetic interactions.

3. Please comment the initial state in the case of 4 lattice-size defect (Fig.1.b) which is unexpected.

4. Fog.5.c is wrong – the distance of the cracks is not 6.

Author Response

(The authors gave the same response as above.)

Reviewer 4 Report

The manuscript presents numerical simulations for cracks defect in magnetostriction of iron thin films. The results are rather interesting but of limited importance. The simulations are focused on two configurations 60x2x2 and 2x60x2 for the same lattice size as parameter (2, 4, 6 and 8) that are rather easy to generate. The reference list is too short and the results are not very well connected to the scientific context.    

Author Response

(The authors gave the same response as above.)

Round 2

Reviewer 1 Report

The corrected manuscript is perceived much better, however, physical errors remain, possibly related to poor knowledge of English. In particular, the explanation provided for "susceptibility" (point 3) describes the total magnetic moment of the sample. "Susceptibility" is a derivative of the magnetic moment by the magnetic field. Appropriate corrections should be made throughout the text. The expressions "the magnetic field size" should be replaced with "the magnetic field induction" or "the magnetic field strength" (lines 117, 119). The explanation (lines 215-219) indicates the imperfection of the program used for modeling. It is necessary to show that under different initial conditions, the results are similar.
In addition, I see that for a more enchanting presentation of the results on the graphs of the field dependences of magnetostrictive stresses, it is desirable to add dependencies for a sample that does not contain defects.
The title of the paragragh 3.1 and 3.2 must be corrected (they must be different).

Author Response

(The authors gave the same response as above.)

Reviewer 3 Report

The Authors positively responded for my suggestions, so, I recommend the paper for publication.

Author Response

Dear reviewer:

Thank you for the thoughtful review and comments on our study.

Reviewer 4 Report

The manuscript was improved.  

Author Response

(The authors gave the same response as above.)
